# *NPM1*-Mutated Patient-Derived AML Cells Are More Vulnerable to Rac1 Inhibition

**DOI:** 10.3390/biomedicines10081881

**Published:** 2022-08-04

**Authors:** Anette Lodvir Hemsing, Kristin Paulsen Rye, Kimberley Joanne Hatfield, Håkon Reikvam

**Affiliations:** 1Department of Medicine, Haukeland University Hospital, 5021 Bergen, Norway; 2Department of Clinical Science, University of Bergen, 5020 Bergen, Norway; 3Department of Immunology and Transfusion Medicine, Haukeland University Hospital, 5021 Bergen, Norway

**Keywords:** Rac1, *NPM1*, signaling molecule, GTPase, AML, cytokines

## Abstract

The prognosis of acute myeloid leukemia (AML) is poor, especially for the elderly population. Targeted therapy with small molecules may be a potential strategy to overcome chemoresistance and improve survival in AML. We investigated the inhibition of the signaling molecule ras-related C3 botulinum toxin substrate 1 (Rac1) in leukemia cells derived from 79 consecutive AML patients, using five Rac1 inhibitors: ZINC69391, ITX3, EHOP-016, 1A-116, and NSC23766. In vitro cell proliferation and apoptosis assays and the assessment of cytokine profiles in culture media were conducted. All five inhibitors had an antiproliferative effect; IC_50_ ranged from 3–24 µM. They induced significant apoptosis and necrosis compared to the untreated controls (*p* < 0.0001) at concentrations around IC_40_ and IC_80_. A high versus an intermediate or low antiproliferative effect was more common in *NPM1*-mutated (*p* = 0.002) and CD34-negative (*p* = 0.008) samples, and when *NPM1* and *FLT3* (*p* = 0.027) were combined. Presence of *NPM1* mutation was associated with reduced viability after treatment with EHOP-016 (*p* = 0.014), ITX3 (*p* = 0.047), and NSC23766 (*p* = 0.003). Several cytokines crucial for leukemogenesis were reduced after culture, with the strongest effects observed for 1A-116 and NSC23766. Our findings suggest potent effects of Rac1 inhibition in primary AML cells and, interestingly, samples harboring *NPM1* mutation seem more vulnerable.

## 1. Introduction

Acute myeloid leukemia (AML) is an aggressive blood cancer with overall low long-time survival despite treatment with intensive chemotherapy and hematopoietic stem cell transplantation. Elderly patients that are not candidates for such intensive therapies often have very limited life expectancies. There is a need for therapies that can alleviate chemoresistance, like targeted therapies toward aberrant signaling molecules. Ras-related C3 botulinum toxin substrate 1 (Rac1) is a signaling molecule localized in the cells’ cytosol and nucleus, involved in several pathways that regulate cell proliferation and migration. It is overexpressed in solid cancers and leukemia [1,2]. The effect of Rac1 inhibition is previously studied in AML cell lines and has demonstrated the capacity to induce apoptosis and improve chemosensitivity [1,3]. A study of proteomic profiles of AML cells with high cytokine release revealed the activation and widespread implication of the Rac1 molecule in different signaling pathways, tightly linked to proteins related to cell migration and adhesion [4]. We sought to study the effect on the proliferation, apoptosis, and altered cytokine profile of five Rac1 inhibitors on a heterogeneous sample of patient-derived AML cells. We hypothesized that the inhibitory effects of the various compounds might be better evaluated and mapped with any possible relation to, e.g., mutational status or cytogenetics.

## 2. Materials and Methods

### 2.1. Primary Human AML Cells

Peripheral blood mononuclear cells were isolated from patients at diagnosis, with at least 80% AML cells among blood leukocytes and a high peripheral white blood cell count (>5.0 × 10^9^/L). The AML cell population was enriched with density gradient separation (Lymphoprep^TM^, Alere Technologies, Oslo, Norway) and included >90–95% primary AML cells. The AML cells were cryopreserved in RPMI-1640 (Merck KGaA, Darmstadt, Germany) with 10% dimethyl sulfoxide (DMSO, Merck KGaA, Darmstadt, Germany) and 20% heat-inactivated fetal bovine serum, stored in liquid nitrogen until thawed and used in experiments. The patients gave written informed consent to participation and collection of biological material in accordance with the Declaration of Helsinki. The regional ethical committee granted approval from local governance (REK III no 060.02 and 059.02, REK Vest 2015/1759, REK Vest 2015.03, REK Vest 2031.06 and REK Vest 2017/305). For our experiments, cells were thawed from 79 consecutive patients. Their characteristics are shown in Table 1.

### 2.2. Pharmacological Agents

Five different Rac1 inhibitors or compounds were purchased: ZINC69391 (Aobious Inc., Gloucester, MA, USA), ITX3 (Merck KGaA, Darmstadt, Germany), EHOP-016 (Merck KGaA, Darmstadt, Germany), 1A-116 (Tocris Bioscience, Bristol, UK), and NSC23766 (Tocris Bioscience, Bristol, UK). Some of their characteristics are listed in Table 2. Their chemical structures are depicted in Appendix A. None of the inhibitors are in clinical use. Initial dissolution was performed in DMSO, then aliquoted in RPMI-1640 and/or StemSpan^TM^ SFEM medium (STEMCELL Technologies, Vancouver, BC, Canada). The final maximal concentration of DMSO was 0.5%.

### 2.3. Proliferation Assay

Patient-derived AML cells for both proliferation and apoptosis assays were cultured in StemSpan^TM^ SFEM medium supplemented with three cytokines: 20 ng/mL final concentration of fms-related tyrosine kinase 3 ligand (FLT3-L), granulocyte-macrophage colony-stimulating factor (GM-CSF), and stem cell factor (SCF) (Peprotech Inc., Cranbury, NJ, USA). Cell proliferation assay was conducted using a ^3^H-Thymidine incorporation assay (PerkinElmer Inc., Waltham, MA, USA) [16]. Patient-derived AML cells were thawed after storage in liquid nitrogen and seeded in a concentration of 5.0 × 10^4^ cells/well, in a flat-bottomed 96 well plate (VWR, Radnor, PA, USA). Vehicle controls (StemSpan^TM^ SFEM medium) and Rac1 inhibitors were added in triplicate to a final volume of 200 µL/well. After six days of incubation at 37 °C and 5% CO_2_, ^3^H-Thymidine was added at a concentration of 37 kBq in 20 µL per well. Cultures were incubated for an additional 20 h before cells were harvested, and nuclear incorporation was determined by liquid scintillation counting. Values of treated cultures were compared with untreated cultures (in medium alone) and reported as ratios for establishing dose-response curves. Detectable proliferation was defined as at least 1000 counts per minute (CPM). The median of triplicate cultures was used in calculations [16].

### 2.4. Apoptosis Assay

The apoptosis assay was conducted with the Annexin V/Propidium Iodine (PI) assay (Nordic BioSite, Täby, Sweden), according to the manufacturer’s protocol. Patient-derived AML cells were thawed after storage in liquid nitrogen and seeded in a concentration of 1.0 × 10^6^ cells/well in a flat-bottomed 24-well plate. Vehicle controls (StemSpan^TM^ SFEM medium) and Rac1 inhibitors were added to a final volume of 1 mL/well. The approximate inhibitory concentration of 40% (IC_40_) and IC_80_ for each Rac1 inhibitor from the pilot proliferation assay were used in this experiment. Cell cultures were incubated for 48 h at 37 °C and 5% CO_2_. According to the manufacturer’s protocol, cell pellets were washed with phosphate-buffered saline and double-stained with Annexin V and PI. Data acquisition was made with a FACSVerse™ flow cytometer (BD Biosciences, San Jose, CA, USA), with a minimum of 10,000 events collected for each sample. Doublet discrimination was performed before gating viable cells. Samples with less than 10% viable cells in controls were discarded.

### 2.5. Cytokine Detection

Cytokine release was measured in the supernatants collected from cell cultures as described in Section 2.4, after 48 h incubation. Supernatants were stored in aliquots at −80 °C until analysis using human magnetic Luminex multiplex assays (R&D Systems, Minneapolis, MN, USA). The results are presented as the measured concentrations without any corrections for cell viability.

### 2.6. Statistical Analysis and Graphical Presentation

Results of the proliferation assay were evaluated with GraphPad Prism v.9.2.0 (GraphPad Software, San Diego, CA, USA) using dose-response curves (non-linear fit). The flow cytometry data were analyzed using FlowJo v.10.7.1 software (BD, Ashland, OR, USA). Statistical analyses were undertaken with GraphPad Prism v.9.2.0 and R v.4.1.2 (R Foundation, Vienna, Austria). Non-parametric t-tests were used. *p*-values < 0.05 were considered statistically significant.

## 3. Results

### 3.1. Rac1 Inhibition Demonstrates Antiproliferative Effects in Primary AML Cells

In an initial pilot proliferation assay, 15 randomly chosen patient samples were cultured to make temporary dose-response curves for each compound. The concentrations for this initial assay were determined based on the previous literature on experiments with tumor cell lines (Table 2). Concentrations tested with few samples often resulted in wide standard deviations, indicating high variability between patient samples. The results are presented in Figure 1.

For each compound, approximate IC resulting in 40% and 80% reduction of proliferation compared to the untreated control in the pilot proliferation assay were used in further experiments with 79 consecutive AML samples. Additionally, new dose-response curves were made with 19 of these samples to have more accurate values for each inhibitor for the future. The new dose-response curves and results from the ^3^H-Thymidine proliferation assay at two fixed concentrations for all the 79 patients (63 with detectable proliferation) are presented in Figure 2A–J.

The IC_50_ values for the Rac1 inhibitors are in the range of 3–24 µM, with EHOP-016 having the lowest value. The results of the ^3^H-Thymidine proliferation assay for all samples at two fixed concentrations indicate that each compound has antiproliferative effects at both concentrations tested and, as expected, some heterogeneity. A broader 2.5–97.5 percentile is seen for the compound ZINC69391, with both high and low concentrations. Table 3 displays the median, error, and range of the two concentrations tested for each compound. For NSC23766, the median results are below the anticipated proliferation, suggesting that both concentrations were too high for the aim, with ratios of 0.60 and 0.20 compared to the untreated control set to 1.

### 3.2. RAC1 Inhibition Demonstrates Proapoptotic Effects in Primary AML Cells

A total of 68 of the 79 patient samples had >10% viable cells and could be assessed with the Annexin V/PI apoptosis assay. An example of the gating strategy used to obtain the results can be seen in Appendix A.

All five compounds significantly induced apoptosis and necrosis at high and low concentrations compared to the untreated controls (Wilcoxon signed-rank test, *p* < 0.0001, Figure 3A). The ratios of apoptosis and necrosis varied, as seen on the y-axis of Figure 3A. ITX3 induced more substantial apoptosis and necrosis than the other compounds; this was also true for the IC_40_. The induction of apoptosis is the main contribution of ITX3, EHOP-016, and NSC23766. Necrosis is the main contribution of the compounds ZINC69391 and 1A-116 (Appendix A). The ratios of apoptosis and necrosis after treatment with the inhibitors ZINC69391 and 1A-116 did not increase considerably with increasing concentrations. However, the ratio of viable cells decreased to become significant (Figure 3B). The ratios of viable cells decreased the most for the highest concentrations of ITX3, EHOP-016, and NSC23766.

### 3.3. Antiproliferative and Proapoptotic Effect Varies between Patients and Is More Frequent in NPM1-Mutated Samples

Hierarchical clustering of the results from the ^3^H-Thymidine proliferation assay was performed in R with Euclidean distance and complete linkage (Figure 4). Only the highest concentrations of the inhibitors were included. The heatmap was subdivided into two main clusters, designating the inhibitors’ overall high versus intermediate or low antiproliferative effect. We used this column in a Fisher’s exact test to evaluate the importance of patient characteristics on the impact of Rac1 inhibition. Fisher’s exact test revealed a significant *p*-value for the presence of *NPM1* mutation (*p* = 0.002) and CD34 negativity (*p* = 0.008) in the cluster with the most potent antiproliferative effect (CD34 positivity defined as >20% positive cells in a patient sample). In the same cluster, the presence of *NPM1* and *FLT3* mutations combined was significant (*p* = 0.027). The evaluation of *FLT3* mutation, FAB classification, cytogenetics, etiology, gender, and age did not reveal significant *p*-values. When analyzing the inhibition of proliferation by NCS23766 alone, significant values were found using Fisher’s exact test, for *NPM1* (*p* = 0.001) and CD34 negativity (*p* = 0.020), *FLT3* (*p* = 0.0005) and for *NPM1* and *FLT3* combined (*p* = 0.011). Moreover, for NSC23766, these analyses were significant when using Kruskal–Wallis with Dunn’s post-test and the Mann–Whitney test comparing groups of patients with or without the mutations (results not shown). For the other compounds analyzed alone, none had a significant association of high antiproliferative effect and presence of mutations with Fisher’s exact test. The Mann–Whitney test found CD34 negativity to be more common with a high antiproliferative effect of the compound ITX3 (*p* = 0.014).

Furthermore, a hierarchical cluster analysis was performed with the Annexin V/PI apoptosis assay results. The results from the highest concentration of each Rac1 inhibitor were used in the heatmap, with Euclidean distance and complete linkage. There was an evident difference in the induction of apoptosis and necrosis by the compounds, as shown in Figure 5. 1A-116 and ZINC69391 induced apoptosis and necrosis in most patient samples in a homogenous manner. ITX3, EHOP-016, and NSC23766 caused a much higher degree of apoptosis and necrosis for at least half of the patient samples. For the most apoptotic and necrotic samples treated with ITX3, CD34 negativity was more common (Fisher’s exact test *p* = 0.024). There were no significant associations with *NPM1* mutation, *FLT3* mutation, FAB classification, or overall antiproliferative effect by the ^3^H-Thymidine proliferation assay.

As expected, NSC23766, EHOP-016, and ITX3 resulted in more profound reduced viability than ZINC69391 and 1A-116 (Figure 6). Again, the presence of *NPM1* mutation was associated with greater reduced viability using Fisher’s exact test. This was true for EHOP-016 (*p* = 0.014) and NSC23766 (*p* = 0.003). For NSC69391 only, CD34 negativity was associated with reduced viability (*p* = 0.006). When comparing the ratio of viability between groups of patients with different characteristics, CD34-negative samples had a significantly lower ratio (reduced viability) when treated with ITX3 (Mann–Whitney test, *p* = 0.043) and NSC23766 (*p* < 0.0001). Similarly, *NPM1*-mutated samples had reduced viability compared to samples with no *NPM1* or *FLT3* mutations when treated with ITX3 (*p* = 0.005) and NSC23766 (*p* = 0.005). Compared to no mutation, samples with combined *FLT3* and *NPM1* mutations had lower viability after treatment with ITX3 (*p* = 0.014) and NSC23766 (*p* = 0.019). There was no significant association with a high or an intermediate or low overall effect by the ^3^H-Thymidine proliferation assay.

### 3.4. Rac1 Inhibition Results in Reduced Release of Cytokines

We investigated the release of different cytokines by measuring their concentrations in the supernatants of 14 AML cell cultures (16 AML samples for one cytokine). The primary release in untreated cultures is summarized in Table 4; it shows a high degree of heterogeneity and is similar to results reported in previous studies [17,18].

Cytokine release after a 48 h culture with the Rac1 inhibitors was then evaluated as a ratio to the untreated control. Medians of the ratios from samples with detectable levels were used in the Wilcoxon signed-rank test for each cytokine (GM-CSF and G-CSF are excluded due to few samples with a detectable release) and for each drug. None of the cytokines had a higher median release than the untreated controls after 48 h of culture. A significantly lower release of several cytokines after Rac1 inhibition, compared to the untreated controls, is displayed in Figure 7. The most significant effects are observed for 1A-116 and NSC23766.

## 4. Discussion

This study explored the in vitro effect of five Rac1 inhibitors, ZIN69391, ITX3, EHOP-016, 1A-116, and NSC23766, on patient-derived AML cells. We demonstrated that Rac1 inhibition leads to the inhibition of cell proliferation, induction of apoptosis and necrosis, and reduced cell viability. These findings are in accordance with previous results on AML cell lines subject to Rac1 inhibition [7]. We established an IC_50_ for each of the Rac1 inhibitors, ranging from 3 to 24 µM, which are generally lower than those found for AML cell lines (Table 2).

The presence of an *NPM1* mutation and/or CD34 negativity of the leukemic blasts was associated with a high overall antiproliferative effect of the Rac1 inhibitors. Similarly, patient-derived samples with combined *NPM1* and *FLT3* mutations had a high overall antiproliferative effect. However, looking at each drug apart, only for NSC23766 could one find an association of a higher antiproliferative effect in *NPM1-* and/or *FLT3*-mutated samples. This can be due to the use of a higher concentration than the other Rac1 inhibitors (Table 3) or the compound itself. CD34 negativity was associated with a high antiproliferative effect after treatment with ITX3.

The heterogeneity in the impact of the Rac1 inhibitors becomes more evident when looking at the heatmaps from the Annexin V/PI apoptosis assay (Figure 5 and Figure 6). ITX3 results in more significant cell death and reduced viability than the other Rac1 inhibitors. We found that 1A-116 and ZINC69391 are more alike in their anti-leukemic effects; this is also true for EHOP-016 and NSC23766. 1A-116 is derived from ZINC69391, and EHOP-016 is derived from NSC23766, making the results plausible. CD34 negativity was associated with more profound apoptosis and necrosis after treatment with ITX3. The Rac1 inhibitors NSC23799, EHOP-016, and ITX3 resulted in greater reduced viability in our apoptosis assay for *NPM1*-mutated and/or CD34-negative samples. Additionally, combined *NPM1-* and *FLT3*-mutated samples were associated with greater reduced viability after treatment with ITX3 and NSC23766. Interestingly, no association between a high antiproliferative effect was observed using the ^3^H-Thymidine proliferation assay and increased cell death or reduced viability using the Annexin V/PI apoptosis assay.

The Rac1 inhibitors significantly reduced cytokine release in the supernatants of the cultured AML cells, with the most potent effects observed for 1A-116 and NSC23766 (Figure 7). Several proinflammatory and proangiogenic cytokines were reduced, implying that cytokines are crucial for leukemogenesis. Five soluble proteins are significantly decreased with inhibition by four or more compounds: CXCL5, Serpin E1, Cystatin C, HGF, and MMP-2. Reduced levels of MMP-2 are interesting since this matrix metalloproteinase is shown to be upregulated in drug-resistant AML and associated with invasive properties of the leukemic cells [19]. Proangiogenic IL-8/CXCL8 and proinflammatory IL-6 are inhibited by both 1A-116 and NSC23766. EHOP-016, 1A-116, and NSC23766 decrease the levels of the antiangiogenic CXCL10. Of note, Xu et al. found different results in a 24 h culture of macrophages with EHOP-016 1 µM; this reduced levels of IL-1β, IL-6, and TNF-α levels in the supernatant of the cultures as well as reduced mRNA levels for the same cytokines [20]. In our study, Rac1 inhibition did not alter the CXCL12 concentration measured in conditioned media. High expression of CXCL12 in the mesenchymal cells and stroma surrounding AML cells in the bone marrow is previously shown to enhance the supporting network and development of AML cells [21]. At least in acute lymphoblastic leukemia, Rac1 activation is observed in response to the CXCL12/CXCR4 axis [1,22].

In normal hematopoiesis, Rac1 is involved in homing and engrafting hematopoietic stem cells. The downstream pathways and regulation of Rac1 are highly intricate. Rac1 is considered inactive when GDP-bound and active when GTP-bound (Figure 8).

This activation is promoted by so-called guanine exchange factors (GEFs) that exchange GDP for GTP molecules. In contrast, guanine-activating proteins (GAPs) deactivate Rac1 by hydrolysis of the bound GTP. When Rac1 is GTP-bound, it binds to different effector proteins that can induce downstream responses in the cell. Rac1 can be attached to GDP dissociation inhibitors (GDIs) in the cytosol to prevent activation and degradation and control the cytosol-to-membrane translocation of the GTPase for activation [23]. GEFs and Rac1 can be activated from upstream receptor tyrosine kinases (e.g., growth factors), the PI3K complex, or integrins [1,24].

Disrupted Rac1 signaling in AML is linked to leukemic infiltration in the bone marrow niche and migration processes of leukemic cells. Active Rac1-GTP in AML promotes pro-survival signals through different pathways and enhances the production of reactive oxygen species (ROS) and inflammation, which renders the surroundings preferable for cancer development [1]. Overexpression of Rac1-GTP can induce disruption of actin filaments in the surrounding stroma and facilitate the reduced functioning of the cell’s DNA damage response and mismatch repair mechanisms [3]. Overexpressed Rac1-GTP can result from upregulated upstream signaling, e.g., *FLT3* mutations in AML cells [6] or mutations in the GEFs or GAPs. Rac1 mutations are not considered relevant patter in AML [1]. Furthermore, Rac1 is more abundant in AML samples with a high constitutive release of extracellular soluble mediators [4].

Wu et al. studied an AML mouse model where Rac1 shRNA silencing made the mice more susceptible to anthracycline chemotherapy [3]. Their results are according to Li et al., who proved reversal of chemoresistance in breast cancer models using Rac1-targeting siRNA together with chemotherapy [25]. Midostaurin (an *FLT3* inhibitor) resistance was alleviated after treatment with the Rac1 inhibitor EHT-1864 in a recent study by Garitano-Trojaola [26]. In that study, it was hypothesized that resistance is led via an increased number of actin filaments, adhesion forces, and increased cell stiffness in the AML cell. In our study, we intended to investigate the inhibitors thought to be more specific for Rac1, thus avoiding, e.g., EHT-1864, which inhibits nucleotide-binding and activation of Rac1 but also the GTPases Rac2 and Rac3 [27], and inhibitors of the GTPase Cdc42, e.g., Aza-1 [28].

However, the specificity of the Rac1 inhibitors used in our study (Table 2) might account for their different effects. The inhibitors bind to various docking sites on the Rac1 molecule, mainly inhibiting interaction with GEFs. ITX3 has not been analyzed to see which other GEFs are inhibited than TrioN. EHOP-016 and NSC23766 inhibit more broadly, including docking sites on the GTPase Rac2, known to be present in hematological tissues and granulocytes, e.g., involved in phagocytosis and cell polarization [29]. NSC23766 inhibits the GEFs Tiam1 and TrioN, but not Vav. EHOP-016 inhibits Vav1, Vav2, and the GTPase Rac3, which is involved in cytoskeletal arrangement in neurological tissues. A recent study identifies RhoG as a target of EHOP-016 and NSC23766 by docking studies [14]. RhoG is a homolog to the Rac GTPases, also involved in cell migration.

Furthermore, the IC_50_s of the inhibitors are in µM, proposed to be too high for further clinical drug development. Rac1 has spread implications in many processes in the cell, and toxicities associated with broad inhibition have been seen. One example is platelet dysfunction in mice treated with NSC23766 and EHT-1864 [1,3,11]. Another challenge is frequent positive and negative feedback loops. Signaling from Rho GTPases might increase when Rac GTPases are inhibited [29,30]. One can speculate if this is what happens with those patient samples that have higher viability after treatment with the Rac1 inhibitors, as shown in Figure 7. A need for a more specific or spatiotemporally Rac1 inhibition has been suggested, like targeting specific GEFs or miRNA [29]. The spatiotemporal regulation of Rac1 and Rho signaling is now better evaluated [31].

Our study revealed that *NPM1*-mutated patient samples were more vulnerable to treatment with the Rac1 inhibitors. They were more likely to show reduced proliferation and viability. The interaction between *NPM1* and Rac1 has been explored by Zoughlami et al. and Navarro-Lérida et al. [23,30]. The latter argue that *NPM1* supports Rac1 activation in the nucleus for the organization of the nuclear membrane and acts as a chaperon to induce the nuclear export of Rac1 to the cytosol. Their experiments showed that the nuclear fraction of Rac1 was about 15% and that the interaction with *NPM1* was facilitated when Rac1 was in an active state. This was in accordance with the results from Zoughlami et al. in HeLa and Jurkat cells. In both studies, activation of Rac1 was unaffected by the knockdown of *NPM1.* However, Navarro-Lérida et al. showed that knockdown of *NPM1* led to nuclear accumulation of fluorescent-tagged Rac1. This also occurred in *NPM1*-mutated AML cells (OCl-AML3 cell line); one could observe nuclear accumulation of fluorescent-tagged Rac1, abnormal nuclear morphology, and protrusions with accumulated Rac1 at the plasma membrane of the cells. In AML, *NPM1* mutations are frequent and lead to the mislocalization and accumulation of *NPM1* in the cytoplasm [32]. One could speculate that such mutations render Rac1 more accessible for the Rac1 inhibitors tested, leading to more significant cell death. This is also relevant for the higher overall antiproliferative effect seen in samples with combined *FLT3* and *NPM1* mutations. Of note, in pull-down experiments, there was no interaction between *NPM1* and Rac1′s homologs RhoA, Rac2, or RhoG [23].

## 5. Conclusions

Our results suggest potent and significant antiproliferative and proapoptotic effects of Rac1 inhibition in patient-derived AML cells, with IC_50_s in the lower µM range. The inhibition is heterogeneous depending on the AML cells’ characteristics, and on the inhibitors and likely their specificity. Interestingly, patient samples harboring the *NPM1* mutation and/or having leukemic blasts with CD34 negativity are significantly more vulnerable to Rac1 inhibition. Our findings can be seen as a framework for further studies evaluating the role of Rac1 and its pharmacological inhibition and alleviation of chemoresistance in AML.

## Figures and Tables

**Figure 1 biomedicines-10-01881-f001:**
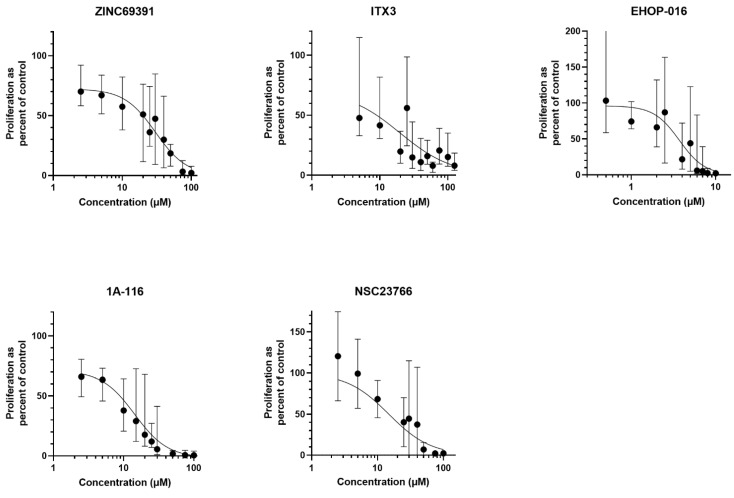
**Pilot dose-response curves.** Dose-response curves (non-linear fit) for each of the five inhibitors were made with a pilot ^3^H-Thymidine proliferation assay after seven days of culture. Proliferation as a ratio to the untreated control on the y-axis, with the untreated control set to 100%. Few patient samples were cultured with each concentration; each dot is the median with a 95% confidence interval (CI) as an error.

**Figure 2 biomedicines-10-01881-f002:**
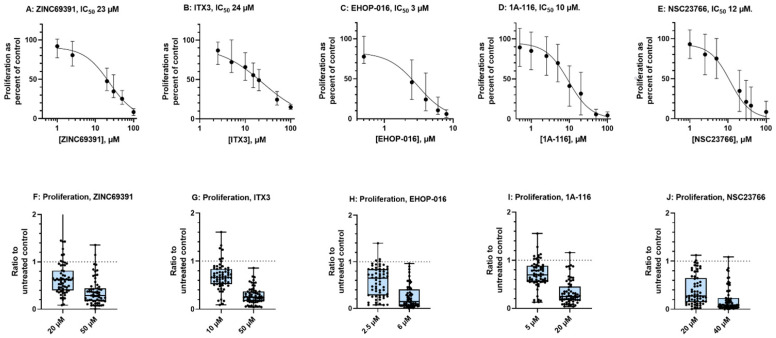
**Results of ^3^H-Thymidine proliferation assay after seven days of culture.** (**A**–**E**): Dose-response curves for each of the five Rac1 inhibitors. The x-axis shows the inhibitor concentration in µM, in log-scale starting at 0.4 µM (except for (**B**) starting at 1 µM). The y-axis represents proliferation as a percent of the untreated control set to 100%. The curves (non-linear fit) are plotted with median values (black dots) and 95% CI as an error. Calculated IC_50_ values and 95% CI error for each compound are as follows: ZINC69391 23 µM (15–29 µM), ITX3 24 µM (15–40 µM), EHOP-016 3 µM (1.8–3.9 µM), 1A-116 10 µM. (8–11 µM), and NSC23766 12 µM (9–15 µM). (**F**–**J**): Proliferation is presented as a ratio to the untreated control (set to 1, dotted line). Each inhibitor is tested at two concentrations for all patient samples, aiming for ratios of 0.60 and 0.20 (lowest and highest concentration, respectively). Results are given as a box plot with whiskers for 2.5 and 97.5 percentiles. Each dot corresponds to an individual patient sample, showing the heterogeneity of the sensitivity toward the compounds.

**Figure 3 biomedicines-10-01881-f003:**
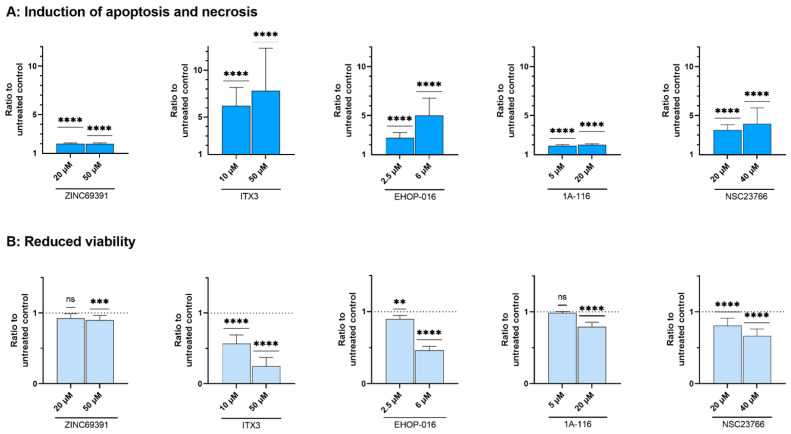
**Results of the Annexin V/PI apoptosis assay.** Flow cytometric assay to evaluate viability, apoptosis, and necrosis after 48 h treatment of 68 patient-derived AML samples with Rac1 inhibitors, assessed with the Annexin V/Propidium Iodine (PI) apoptosis assay. The results are presented as a ratio to the untreated control set to 1 (dotted line in (**B**)). Columns represent medians with 95% CI as an error. For each Rac1 inhibitor, the graphs show (**A**) the induction of apoptosis and necrosis combined and (**B**) the cell viability with two different concentrations of the inhibitors. Wilcoxon signed-rank test was performed for evaluation against the untreated control set to 1. ns: non-significant. ** *p* < 0.01, *** *p* < 0.001, **** *p* < 0.0001.

**Figure 4 biomedicines-10-01881-f004:**
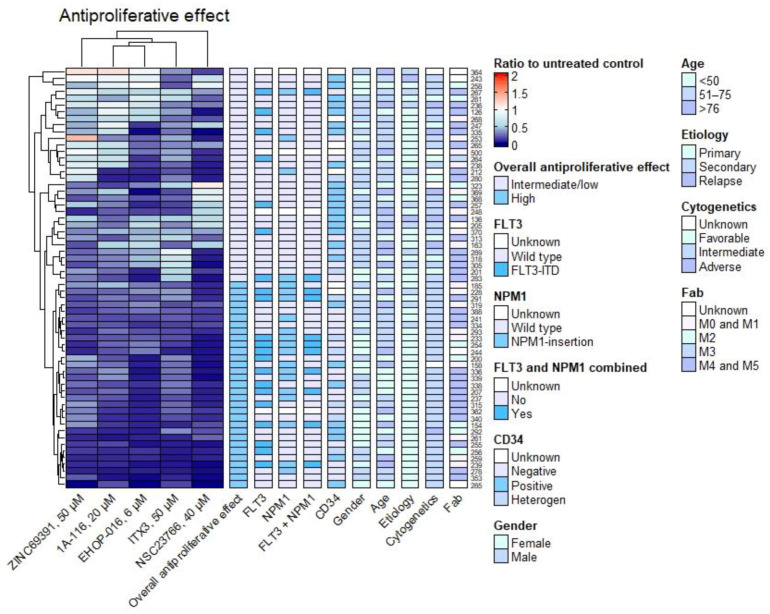
**Rac1 inhibition has an antiproliferative effect.** Proliferation is shown as the ratio of treated cells to the untreated control after seven days of culture. Each row is a patient sample, and each column in the heatmap is a Rac1 inhibitor. The heatmap is made with Euclidean clustering and complete linkage. Patient mutational status and CD34 status of the leukemic blasts are indicated on the right side of the heatmap, along with patient characteristics. The high overall antiproliferative effect of the inhibitors was more common in *NPM1*-mutated (*p* = 0.002) and CD34-negative (*p* = 0.008) samples, as well as samples with combined *NPM1* and *FLT3* mutations (*p* = 0.027) using Fisher’s exact test.

**Figure 5 biomedicines-10-01881-f005:**
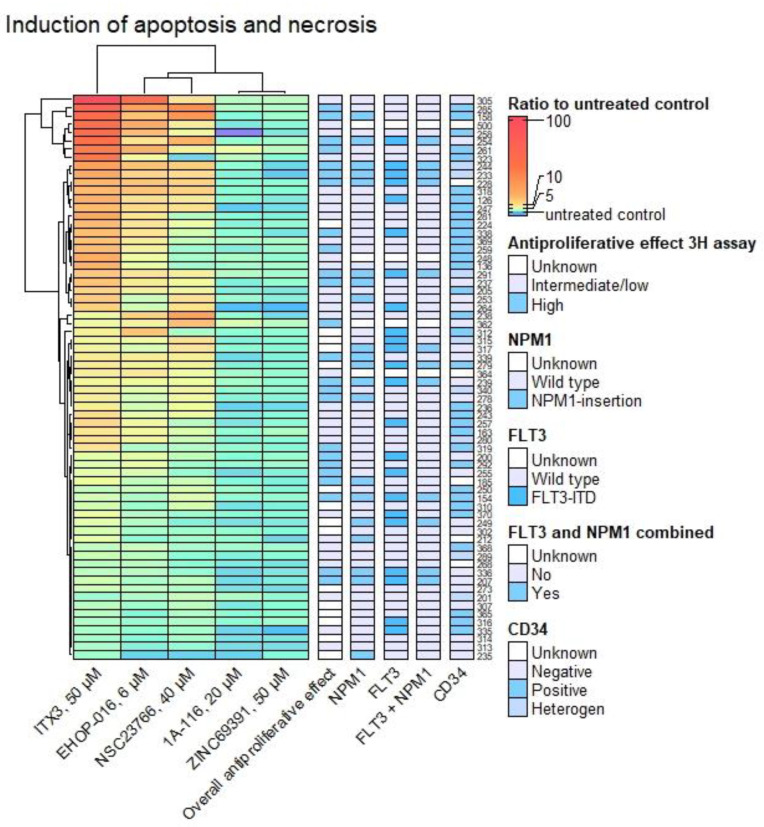
**Rac1 inhibition induced apoptosis and necrosis in AML cells.** Heatmap of apoptosis and necrosis combined by the Annexin V/PI apoptosis assay after 48 h of culture, as a ratio to the untreated control set to 1. Blue and purple colors indicate a lower ratio of apoptosis and necrosis than the control. Green indicates a slightly higher ratio than the control; yellow, orange, and red colors, even greater. Each row is a patient sample; each column is a Rac1-inhibitor, with the highest concentration tested. The columns on the right side of the heatmap indicate *NPM1* and *FLT3* mutational status and CD34 expression for each patient sample. The column “Overall antiproliferative effect” indicates whether the sample had an overall high or an intermediate or low antiproliferative effect of the compounds by the ^3^H-Thymidine proliferation assay. A Fisher’s exact test was applied by dividing the compounds’ overall heatmap, and separate heatmaps for each compound (not shown), into two groups that were analyzed with the available characteristics. More profound apoptosis and necrosis were common in CD34-negative samples only for ITX3 (*p* = 0.024). No other significant results were found when looking for associations with a high degree of apoptosis and necrosis.

**Figure 6 biomedicines-10-01881-f006:**
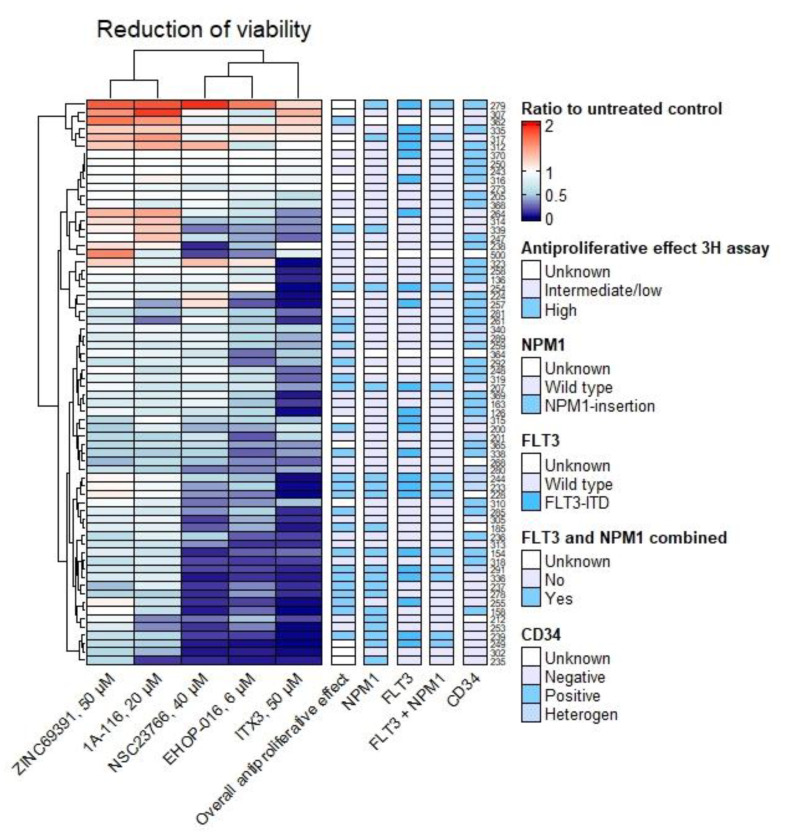
**Rac1 inhibition reduced viability.** Heatmap of reduced viability by the Annexin V/PI apoptosis assay after 48 h of culture, as a ratio to the untreated control. Each row is a patient sample; each column is a Rac1-inhibitor. The columns on the right side of the heatmap indicate different characteristics of each patient sample. The column “Overall antiproliferative effect” indicates whether the sample had an overall high or intermediate/low antiproliferative effect of the compounds by the ^3^H-Thymidine proliferation assay. A more profound reduction of viability was common in the *NPM1*-mutated samples when treated with EHOP-016 (*p* = 0.014) and NSC23766 (*p* = 0.003) using Fisher’s exact test, and with ITX3 (*p* = 0.005) and NSC23766 (*p* = 0.005) using the Mann–Whitney test (comparing groups of patients with and without the mutation). Samples with combined *FLT3* and *NPM1* mutations compared to no mutation had lower viability after treatment with ITX3 (*p* = 0.014) and NSC23766 (*p* = 0.019) using the Mann–Whitney test. CD34-negative samples had significantly more reduced viability when treated with ITX3 (*p* = 0.043) and NSC23766 (*p* < 0.0001) using the Mann–Whitney test, for NSC23766 also using Fisher’s exact test (*p* = 0.006).

**Figure 7 biomedicines-10-01881-f007:**
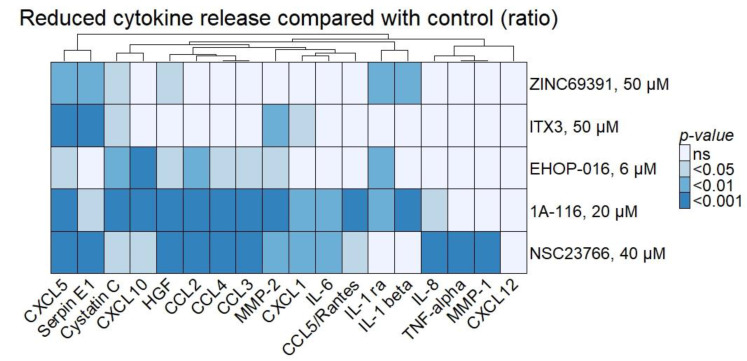
**Rac1 inhibition reduced cytokine release in the culture medium.** The Rac1 inhibitors significantly reduced several proinflammatory cytokines released in the culture medium after 48 h. Each sample was evaluated as a ratio to the untreated control for each cytokine and inhibitor. The median of 10–16 patient samples was assessed with Wilcoxon signed-rank test against the control set to 1. The resulting *p*-values are presented in the heatmap with Euclidean clustering and complete linkage. The sample size of the cytokine analysis was too small for the importance of *NPM1* mutations or other characteristics to be considered. ns: non-significant.

**Figure 8 biomedicines-10-01881-f008:**
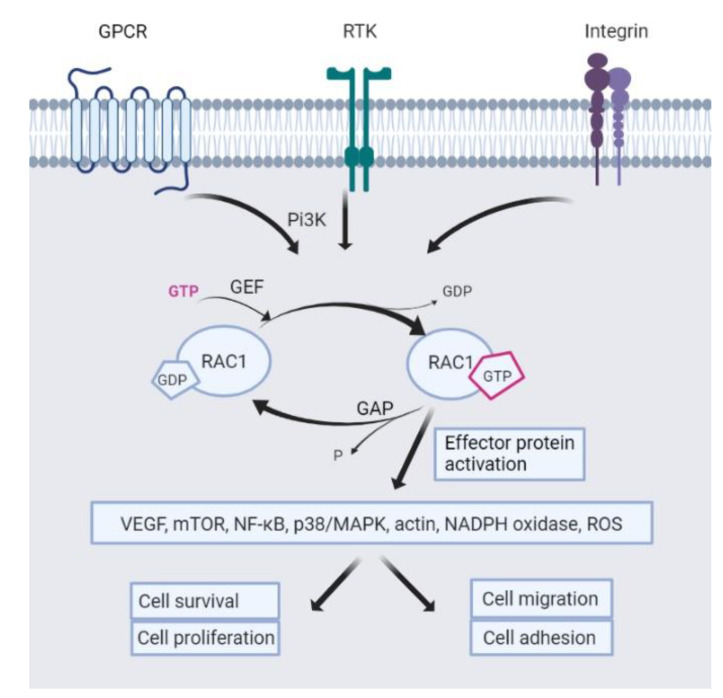
**Rac1 regulation.** Depiction of activation and deactivation of the GTPase Rac1 by GEFs and guanine-activating proteins (GAPs). Downstream effector protein activation results in the activation of various signaling pathways, leading to cell survival.

**Table 1 biomedicines-10-01881-t001:** **Patient characteristics.** Demographical and clinical characteristics of the 79 acute myeloid leukemia (AML) patients included in the study. Cytogenetic risk classification is based on the European Leukemia Net classification of 2017 [5]. * Intermediate cytogenetics include normal cytogenetics; 36 samples of the 56.

PATIENT CHARACTERISTICS		OBSERVATIONS
Demographic Data and Disease History		
Gender (numbers)	Female/male	33/46
Age, years (median, range)		59 (17–90)
History (number, percent)	De novo	61 (77)
	Secondary	16 (20)
	Relapse	2 (3)
Hematology (mean, range)		
Hemoglobin, g/dL		9.6 (3.8–15.4)
Platelets, ×10^9^/L		76 (11–258)
White blood cells, ×10^9^/L		71.8 (5–660)
AML cell differentiation (number, percent)		
FAB	M0-1	18 (23)
	M2	10 (13)
	M3	1 (1)
	M4-5	45 (57)
	Unclassified	5 (6)
Genetic abnormalities (number, percent)		
Cytogenetics	Favorable	11 (14)
	Intermediate *	56 (71)
	Adverse	7 (9)
	Unknown	5 (6)
*FLT3*	Wild type	48 (61)
	ITD/TKD	25/1 (33)
	Unknown	5 (6)
*NPM1*	Wild type	49 (62)
	Insertion	25 (32)
	Unknown	5 (6)

**Table 2 biomedicines-10-01881-t002:** **Characteristics of the ras-related C3 botulinum toxin substrate 1 (Rac1) inhibitors used in this study.** The inhibitory concentration of 50% (IC_50_) against cell lines and specificity was found with a search for the inhibitors on PubMed. The specificity was, in general, demonstrated by Western blot studies. Tiam1, Dock180, TrioN, Vav1, Vav2 and Vav3, Dbl, and P-Rex1 are guanine exchange factors (GEFs) that activate Rac1. Rac1, Rac2, Rac3, Cdc42, and RhoG are GTPases.

Inhibitor	ZINC69391	ITX3	EHOP-016	1A-116	NSC23766
Chemical formula	C_14_H_14_F_3_N_5_	C_22_H_17_N_3_OS	C_25_H_30_N_6_O	C_16_H_16_F_3_N_3_	C_24_H_35_N_7.3_HCl
Molecular weight (g/mol)	309.29	371.45	430.55	307.31	530.9
CAS number	303094-67-9	347323-96-0	1380432-32-5	1430208-73-3	1177865-17-6
IC_50_ against cell lines	IC_50_ 41.7–54.1 µM, AML cell lines U937, HL60 and KG1a [6]. IC_50_ 31–61 µM, breast cancer cell lines MCF7, F3II and MDA-MB-231 [7].	Unknown	IC_50_ 1.1–3 µM, breast cancer cell lines MDA-MB-435 and MDA-MB-231 [8]. IC_50_ 4.3–9.1 µM, lung cancer cell lines Calu1, A549 and HOP62 [9].	IC_50_ 26–63 µM, AML cell lines U937, HL60 and KG1a [6]. IC_50_ 4–21 µM, breast cancer cell lines F3II and MDA-MB-231 [7].	IC_50_ 25–100 µM, AML cell lines HL60 and KG1a [10]. IC_50_ 140uM, breast cancer cell line F3II [7].
Specificity	Inhibit Tiam1 and Dock180 (=Dock1) [7,11].	Inhibit TrioN. No effect on Tiam1 and Vav2 [12].	Inhibit Rac2, Rac3, Vav1, Vav2, RhoG, and Cdc42 at higher concentrations of 5–10 µM. No effect on Tiam-1 and TrioN [8,13,14].	Inhibit P-Rex1, Vav1, Vav2, Vav3, Tiam1 and Dbl [7,15].	Inhibit TrioN, Tiam1, Rac2, RhoG, TrioN and Tiam-1. No effect on Vav [1,8,14].

**Table 3 biomedicines-10-01881-t003:** **Tabular view of the proliferation assay.** The median proliferation determined by the ^3^H-Thymidine assay of 63 patient-derived samples was assessed as a ratio to the untreated control set to 1. We anticipated that the highest concentration would result in a ratio of approximately 0.20 and the lowest concentration in a ratio of approximately 0.60. The actual median is presented in the table, with 95% CI as an error. Range from minimum to maximum proliferation, as well as mean values, are indicated.

Inhibitor	ZINC69391	ITX3	EHOP-016	1A-116	NSC23766
Highest concentration (µM)	50	50	6	20	40
Median	0.29	0.25	0.15	0.25	0.09
95% CI of median	0.20–0.28	0.22–0.32	0.08–0.27	0.20–0.31	0.07–0.15
Min-Max	0.004–1.40	0.04–0.86	0.002–0.97	0.04–1.16	0.001–1.09
Mean	0.36	0.29	0.25	0.33	0.21
Lowest concentration (µM)	20	10	2.5	5	20
Median	0.62	0.66	0.65	0.69	0.28
95% CI of median	0.49–0.65	0.61–0.72	0.43–0.73	0.60–0.79	0.22–0.41
Min-Max	0.09–2.50	0.09–1.61	0.07–1.40	0.13–1.56	0.004–1.12
Mean	0.66	0.68	0.58	0.71	0.40

**Table 4 biomedicines-10-01881-t004:** **Cytokine release.** Cytokine release in culture supernatants from 14 consecutive AML samples (16 samples for IL-8) after 48 h of culture without Rac1 inhibitors.

Cytokine	Number of Patients with Detectable Levels	Median Concentration and Range (pg/mL)
CCL2	13 of 14	4749 (11–19,733)
CCL3	14 of 14	5662 (32–82,404)
CCL4	14 of 14	2076 (93–33,507)
CCL5	14 of 14	273 (5–1338)
CXCL1	10 of 14	2113 (64–39,153)
CXCL5	14 of 14	1377 (93–89,755)
CXCL10	13 of 14	34 (4–1333)
CXCL12	14 of 14	366 (291–471)
IL-1 RA	10 of 14	7821 (56–28,546)
IL-8	16 of 16	5207 (146–268,468)
IL-1β	11 of 14	47 (2–1721)
IL-6	10 of 14	63 (2–736)
G-CSF	6 of 14	82 (4–32,133)
HGF	14 of 14	84 (8–704)
TNF-α	12 of 14	70 (3–1233)
GM-CSF	7 of 14	1820 (1310–4036)
Cystatin C	14 of 14	7239 (1075–10,621)
Serpin-E1	14 of 14	1477 (73–9619)
MMP-1	12 of 14	294 (12–16,253)
MMP-2	14 of 14	3013 (213–8105)

## Data Availability

The data presented in this study are available on request from the corresponding author. The data are not publicly available due to privacy. The hierarchical clustering shown in Figure 4, Figure 5 and Figure 6 was done in R with the package Complexheatmap [33]. Figure 8 was made with Biorender.com on the 28 June 2022.

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
