# Peer review of "NPM1-Mutated Patient-Derived AML Cells Are More Vulnerable to Rac1 Inhibition"

_biomedicines, 2022, doi:10.3390/biomedicines10081881_

Round 1

Reviewer 1 Report

The manuscript titled “NPM1 mutated patient-derived AML cells are more vulnerable to Rac1 inhibition” explored the five different Rac1 inhibitors’ roles in AML cells which are derived from patients’ samples. According to the various IC50 of individual inhibitors, authors compared cell proliferative ability, apoptosis, and necrosis ability and found out NPM1 mutated cells are more sensitive to Rac1 blockage. Overall, all the methods are reasonable and the manuscript is interesting. My major concerns are blow:

1. How long did the five Rac1 inhibitors treated with patients’ samples? Please illustrate in figure 1 and figure 2 legend.

2. I suggest it’s better to put figure 3 in the supplementary because the gating strategy is usually common in the apoptotic study.

3.  In figure 4, both A and B, please let the characters of different inhibitors in the middle of the bottom.

4. In page 11 line 266, figure 7 legend is “Rac1 inhibition r viability”, is this correct? What is “r”?

5. Are the five inhibitors already clinic used? What’s the recently progressive of these inhibitors in clinical?

Author Response

Dear Reviewer 1, 

Thank you for your relevant comments on the manuscript.

  1. How long did the five Rac1 inhibitors treated with patients’ samples? Please illustrate in figure 1 and figure 2 legend.

Thank you for observing this lacking information in figure 2 legend. The information has now been included in the title to highlight that this applies to all figures 2A-J. The time of culture is described in figure 1 legend.

  1. I suggest it’s better to put figure 3 in the supplementary because the gating strategy is usually common in the apoptotic study.

We agree that this is reasonable. Figure 3 will now be assigned as a supplemental figure. Hence, the numbering of the figures will be altered in the revised manuscript.

  1.  In figure 4, both A and B, please let the characters of different inhibitors in the middle of the bottom.

This has been addressed, and a new file for figure 4 has been uploaded.

  1. In page 11 line 266, figure 7 legend is “Rac1 inhibition r viability”, is this correct? What is “r”?

Thank you for highlighting a missing word. The title of Figure 7 should read “Rac1 inhibition reduce viability”. This is now corrected in the manuscript.

  1. Are the five inhibitors already clinic used? What’s the recently progressive of these inhibitors in clinical?

These inhibitors are not in clinical use or clinical trials; this is now mentioned in the method section 2.2. Some of them have been studied in animal models.

Reviewer 2 Report

This research manuscript is interesting and well written. The length of the manuscript is accurate and the data is properly presented and discussed. Some stylistic issues should be corrected by the authors in a revised version of the manuscript. Thus, I recommend acceptance after minor revision.

Numbers in quotations such as ´´IC50´´ and ´´CO2´´ should be written in subscript (IC50 and CO2) throughout the manuscript.

Table 2: Please specify the cell lines in the table or as footnote. The molecular weight values are on different lines, please make conform. ´´Chemical structure´´ is wrong here because it shows no structures in the table, replace it by ´´Chemical formula´´.

I suggest that the authors provide the chemical structures of the used Rac1 inhibitors in a separate figure.

Figure 9: Please correct ´´NF-kB´´ and ´´ACTIN´´ (replace by ´´NF-κB´´ and ´´actin´´).

Author Response

Dear Reviewer 2,

Thank you for your relevant comments on the manuscript.

Numbers in quotations such as ´´IC50´´ and ´´CO2´´ should be written in subscript (IC50 and CO2) throughout the manuscript.

This has been corrected.

Table 2: Please specify the cell lines in the table or as footnote. The molecular weight values are on different lines, please make conform. ´´Chemical structure´´ is wrong here because it shows no structures in the table, replace it by ´´Chemical formula´´.

Thank you for the relevant and constructive feedback on Table 2. Specification of the cell lines is now included in the table. “Chemical structure” has been changed to “chemical formula.”

I suggest that the authors provide the chemical structures of the used Rac1 inhibitors in a separate figure.

A supplementary figure with the chemical structure of each inhibitor is made. As a consequence, and due to reviewer 1 suggesting figure 3 to be put in the supplementary, the numbering of the figures is altered in the revised manuscript.

Figure 9: Please correct ´´NF-kB´´ and ´´ACTIN´´ (replace by ´´NF-κB´´ and ´´actin´´).

This has been corrected in Figure 9.

Round 2

Reviewer 1 Report

All my questions are well answered. I suggest it be accepted by the journal.